# Identifying opioid agonist treatment prescriber networks from health administrative data: A validation study

Megan Kurz[1,2], Mark Tatangelo[3,4], Kristen A. Morin[3,4,5], Michelle Zanette[1], Emanuel Krebs[6], David C. Marsh◍[3,4,5], Bohdan Nosyk◍[1,2]*

**1** Centre for Advancing Health Outcomes, Vancouver, BC, Canada, **2** Faculty of Health Sciences, Simon Fraser University, Burnaby, BC, Canada, **3** Health Sciences North, Sudbury, Ontario, Canada, **4** ICES North, Sudbury, Ontario, Canada, **5** Northern Ontario School of Medicine University, Sudbury, Ontario, Canada, **6** BC Cancer Research Institute, Vancouver, BC, Canada

* bnosyk@sfu.ca

## Abstract

### Background

Given the growth of collaborative care strategies for people with opioid use disorder and the changing composition of the illicit drug supply, there is a need to identify and analyze clinic-level outcomes for centers prescribing opioid agonist treatment (OAT). We aimed to determine and validate whether prescriber networks, constructed with administrative data, can successfully identify distinct clinical practice facilities in Ontario, Canada.

### Methods

We executed a retrospective population-based cohort study using OAT prescription records from the Canadian Addiction Treatment Centres in Ontario, Canada between 01/01/2013 and 12/31/2020. Social network analysis was utilized to create networks with connections between physicians based on their shared OAT clients. We defined connections two different ways, by including the number of clients shared or a relative threshold on the percentage of shared OAT clients per physician. Clinics were identified using modularity maximization, with sensitivity analyses applying Louvain, Walktrap, and Label Propagation algorithms. Concordance between network-identified facilities and the (gold standard) de-identified facility-level IDs was assessed using overall, positive and negative agreement, sensitivity, specificity, positive predictive value (PPV) and negative predictive value (NPV).

### Results

From 144 physicians at 105 clinics with 32,842 OAT clients, we assessed 250 different versions of the created networks. The three different detection algorithms

**Data availability statement:** The data underlying the results presented in this study cannot be shared in a public repository due to privacy restrictions, but are available from the Canadian Addiction Treatment Centres (CATC; https://canatc.ca/) upon reasonable request. Requests for access to the data can be directed to April Gamache, Chief Executive Officer of CATC (email: agamache@canatc.ca). The CATC is located at 175 Commerce Valley Drive West, Suite 300, Markham, Ontario, L3T 7P6, and can be reached toll-free at 1-877-937-2282. Further details on the data are available in the cohort profile: Morin KA, Tatangelo M, Marsh D. Canadian Addiction Treatment Centre (CATC) opioid agonist treatment cohort in Ontario, Canada. BMJ Open 2024;14:e080790. doi: 10.1136/bmjopen-2023-080790. All the statistical codes used for the analysis are available in the following repository: https://github.com/HERU-modeling/Network-Validation/.

**Funding:** This work was funded by the Canadian Institutes of Health Research award number no. PJT-190265 and National Institutes of Health/National Institute on Drug Abuse award no. R01-DA050629. The funders had no role in study design, data collection and analysis, decision to publish, or preparation of the manuscript.

**Competing interests:** DM previously held the position of Chief Medical Director at CATC ending in June 2022 and maintains the role of OAT provider. DM had no ownership stake in the CATC as a stipendiary employee. We do not forsee any conflict of interest as findings will be made freely available to the public and the CATC, and neither the Universities nor CATC can prevent publication and dissemination of knowledge. There are no patents or products related to this submission. There are no competing interests that would alter myself and my co-authors adherence to PLOS ONE policies for sharing data and materials.

had wide variation in concordance, with ranges on sensitivity from 0.02 to 0.88 and PPV from 0.06 to 0.97. The optimal result, derived from the modularity maximization method, achieved high specificity (0.98, 95% CI: 0.98, 0.98) and NPV (0.98, 95% CI: 0.97, 0.98) and moderate PPV (0.54, 95% CI: 0.52, 0.57) and sensitivity (0.45, 95% CI: 0.43, 0.47). This scenario had an overall agreement of 0.96, negative agreement of 0.98, and positive agreement of 0.49.

## Conclusions

Social network analysis can be used to identify clinics prescribing OAT in the absence of clinic-level identifiers, thus facilitating construction and comparison of clinic-level caseloads and treatment outcomes.

---

## Introduction

Given the growing interest in the development of collaborative and continuing care strategies for people with opioid use disorder (PWOUD) [1–4], there is considerable clinical and scientific importance associated with validating prescriber networks and subsequently understanding the benefit of physician networks on treatment retention. As defined by the Agency for Healthcare Research and Quality, care coordination is the "deliberate organization of patient care activities between two or more participants involved in a patient's care to facilitate appropriate delivery of health care services" [5]. Opioid agonist treatment (OAT) is the standard treatment for opioid use disorder and improving treatment retention has been a key component in public health strategies to reduce the risk of overdose and subsequent mortality among PWOUD [6–10]. While structural and individual-level characteristics have an impact on retention in OAT [11], clinic-level characteristics, such as receiving care at a specialized treatment center or hospital [12], are also important determinants of OAT retention [11,13–17] which are directly amenable to systemic intervention.

Within Canada, 80% of all Canadians receiving OAT reside in Ontario or British Columbia (BC) [18]. The largest treatment provider organization in Ontario is the Canadian Addiction Treatment Centre (CATC), which operates outpatient and inpatient treatment, with specialized pharmacies [19]. In BC, OAT is provided through specialized treatment centres, such as rapid access to addiction care clinics, community health centers and physicians' private offices. Like Ontario, BC boasts comprehensive linked health administrative data, however clinic-level identifiers are not available within these databases [20], thus limiting the opportunity for clinic-level comparisons in practice patterns and outcomes. Federal and Provincial governments have proposed increasing connections to care and collaborative care practices as interventions to improve outcomes for individuals with substance use disorders [21–23]. The Canadian Centre on Substance Abuse recommends single assessment processes, shared medical records and centralized access points as mechanisms to work towards collaborative care [24].

Evaluating the performance of collaborative care and speciality clinics can be labour-intensive and thus cost-prohibitive when conducted through observations of practices, qualitative interviews and surveys. The use of administrative data may be feasible for routine, ongoing evaluation but many datasets, lack or suppress facility-level identifiers, including only de-identified provider information. Two previous studies conducted in the United States demonstrated that patient-sharing, identified using administrative data, is an informative measure for determining relationships between physicians [25,26]. More recently, a study used Medicare data to identify shared care between primary care providers [27]. Given the case mix complexity and potential need for multidisciplinary care for people presenting with OUD, these methods require validation within this population and care context to ensure accuracy. However, no studies exist to validate the use of social network analysis to measure relationships among OAT physicians, and further identify clinics when administrative data lacks the identifying information on facilities or survey data is unavailable. The inclusion of de-identified clinic identifiers in Ontario's CATC data presents an opportunity to fill this gap.

Social network analysis can be further applied to identify the impact of these networks on patient outcomes and health care system performance through the evaluation of care coordination [5,28,29]. While care coordination is challenging to measure through administrative claims data, "care density" is a surrogate measure which suggests that indirectly, through having visits with the same patients, physicians develop relationships that result in opportunities for direct communication and information sharing that may lower barriers to care coordination and ultimately lower costs [5]. Other applications of social network analysis include improving the implementation of new policies or clinical guidelines through network interventions that utilize the theory of diffusion of innovation by identifying and targeting individuals or groups for behaviour change [30,31]. Validating the identification of clinics via administrative data provides the groundwork to identify and target groups of clinicians on a clinic-by-clinic basis for a segmentation style network intervention [30] while utilizing the lower evaluation cost associated with administrative data.

We aimed to determine whether prescriber networks, constructed via de-identified physician IDs in daily drug dispensation records, can successfully identify distinct clinical practices. We employ electronic medical record data from the largest network of physicians providing OAT in Canada (a group of 105 OAT clinics in Ontario, Canada) to assess the concordance between constructed prescriber networks and actual relationships between physicians.

## Methods

### Study design

To validate the physician communities identified through the network analysis we considered whether physician pairs treated any clients at the same clinic as the gold standard and compared with the physician pairs' community (network identified clinic) that was assigned through our analysis. The pairing was concordant if they prescribed to clients at the same clinic and were assigned to the same community, or if they did not care for clients at the same clinic and were in different communities.

### Study population & data sources

Electronic medical records from CATC were used to create a retrospective population-based cohort. CATC is a network of 105 treatment centers in Ontario, Canada that captures OAT prescriptions, clinic identifiers, prescriber identifiers and client identifiers [19]. The cohort was comprised of all physicians with a prescription from a clinic within the CATC network between January 2013 and December 2020. We defined physician-clinic connection as any OAT prescriber with a record in a given clinic. Physician- and clinic-client connection was defined as any OAT prescription between the physician and/or clinic and the OAT client.

### Data collection and ethics approval

Data were obtained on February 1st, 2021, from CATC. The data were de-identified and all individual-level characteristics were removed prior to analysis. We received approval from the Laurentian University Research and Ethics Board for this study (Approval number 6020852).

## Prescriber Network construction

We used de-identified clinic-level identifiers in the CATC dataset, based on prescription records of the prescribing physician, as our gold standard indication of a prescriber's clinic membership. Social network analysis was utilized to create a network with connections between physicians based on their shared OAT clients. A network is defined as a set of 'actors' (prescribers) with 'relational ties' (shared clients) [32]. As the network was constructed on the prescriber-level, a 'connection' between two prescribers entailed prescription to at least one shared OAT client. Within the network, the connections were weighted to represent the total number of shared OAT clients, such that the greater the weighted tie, the more clients were shared between prescribers. Sub-network groups, commonly referred to as 'communities' in the network literature, were identified based on statistically non-random connections using modularity maximization [33]. This involved assigning prescribers to disjoint communities through comparing the differences between the true number of shared clients and the number of expected by chance, given each pre-scriber's client loads. We applied the modularity maximization procedure recursively to identify smaller sub-group more likely to be representative of clinics, as seen in Stein et al (2017) [34].

## Statistical analysis

Concordance was assessed through several measures, including percent overall agreement, positive agreement, negative agreement and Gwet's agreement coefficient ($AC_1$). Gwet's $AC_1$ was chosen over Cohen's Kappa as it is more reliable in scenarios with low prevalence and imbalanced marginal probabilities [35]. We also calculated the sensitivity (probability for a physician pair to be grouped in a network-identified community together when they are attached to the same clinic), specificity (probability for a physician pair to be in separate network-identified communities when they do not practice at the same clinics), positive predictive value (PPV; probability of a physician pair practicing at the same clinic when they are grouped in a network-identified community) and negative predictive value (NPV; probability of a physician pair practicing at different clinics when they are in different network-identified communities). Following previous studies that validated network-identified clinics [27,36] we also estimated recall (the percentage of physicians in the network-identified clinic that are in the true clinic), 1-purity (percentage of physicians in the network-identified clinic that are not in the true clinic) and the F-measure (the harmonic mean of purity and recall values of each network-identified clinic, where a maximum value of 1 indicates perfect agreement) for the network-identified clinics.

Several sensitivity analyses were executed to ensure the results were robust. We applied several alternative defi-nitions of ties in the network, including different minimum numbers of shared clients (absolute thresholds), retain-ing only the top 20–80 percent of a physicians weighted ties (based on shared clients, relative threshold), and ties defined on shared OAT clients treated during the same OAT episode (defined as continuous OAT without treatment gaps lasting at least 5 days for methadone or 6 days of buprenorphine/naloxone [37]). We also applied several dif-ferent definitions of physician-clinic connection, were we defined a physician-clinic connection if the clinic was where the physician prescribed to the majority of their OAT clients or prescribed to more than 15, 30 or 45 percent of their total OAT clients. In addition to the modularity maximization method to identify disjoint communities, we compared the Walktrap [38] and Louvain [39] methods to test different identification algorithms, as they have been used in other community detection studies [26,27]. We also compared the Label Propagation [40] algorithm, although it has not been used in community detection studies among prescribers. Lastly, we used several different recursive itera-tions of the modularity maximization procedure to define the smaller communities. To identify which algorithm was best for detection of communities, we prioritized sensitivity and positive predictive value, while expecting that speci-ficity and negative predictive value should be high due to the large amount of true physician-pairs that do not care for clients at the same clinics.

## Results

### Characteristics of physicians, clinics and identified facilities

We included 144 physicians treating 32,842 OAT clients at 105 clinics over the 7-year study period to assess 350 different versions of the created networks (S1 Table). Clinics were identified on an annual basis. Between January 1st 2020 and December 31st 2020 there were 59 CATC clinics that treated OAT clients, and we identified 36 communities where physician-clinic connection was defined as at least 15% of the physician's client load (Table 1, S2 Table). Among these communities, recall (the number of physicians identified in the communities that were also in the administrative identified clinic) had a median of 50%, 1-purity (the number of physicians identified in communities that were not in the administrative identified clinic) had a median of 0.

Among the 36 communities, we presented three clinics and their matching network-identified practice as an example visualization (Fig 1). Clinic #25 was identified in the networks with no additional or missing physicians, however as the two overlapping physicians' sub-communities had no clients shared, they were identified as two separate distinct clusters. Clinic #154 in the network-identified clinics had missed the inclusion of three physicians, however, all missing physicians (identified as H, G, and F) had small client loads (one to four clients). Finally, Clinic #149 missed the inclusion of two physicians, with "H" having a small client load (4 clients, all treated at different clinics), and "I" having a large client load, but more of their client load at a different clinic.

### Concordance statistics for network-identified clinics

When varying the detection algorithms and physician-clinic connection definitions, with ties defined under the relative threshold at 80%, sensitivity estimates ranged from 0.02 to 0.93 and PPV estimates ranged from 0.04 to 0.98 (Fig 2). Specificity estimates ranged from 0.49 to 1.00 and NPV estimates ranged from 0.44 to 1.00. Results from the Walktrap, Louvain, and Label Propagation algorithms had lower specificity and PPV compared to the modularity maximization algorithm. The Louvain algorithm had slightly higher estimates among all categories than Walktrap, except when physician-clinic connection was defined as any OAT client treated through the clinic. Label Propagation had the highest sensitivity, with the drawbacks of the lowest PPV and specificity. Among the modularity maximization algorithms, the fourth iterations' 0.03 increase in PPV resulted in a 0.12 reduction in sensitivity. Altering the definition of a true physician-clinic connection from the percentage of a physician's client load to any clients treated at the clinic reduced sensitivity to near zero, and NPV to near 0.50 (Fig 2, S3 Table).

**Table 1. Summary of characteristics for network-identified clinics and administrative identified clinics from 2020.**

| | Network identified communities (n = 36) | Administrative identified clinics (n = 59) |
|---|---|---|
| | Median (Q1 - Q3) | |
| No. physicians | 2 (1- 4) | 3 (2 - 4) |
| No. clients | 137 (4 - 726) | 189 (93 - 326) |
| Physicians' client load | 86 (3 - 253) | 93 (26 - 153) |
| No. ties | 21 (7 - 46) | 19 (8 - 26) |
| Recall (% of physicians in the network-identified clinic that are in the true clinic) | 50.0 (23.8 - 89.3) | ---------- |
| 1-purity (% of physician in the network-identified clinic that are not in the true clinic) | 0 (0 - 18.8) | ---------- |
| F-measure (harmonic mean of purity and recall values; maximum value of 1 indicates perfect agreement). | 0.55 (0.33–0.77) | ---------- |

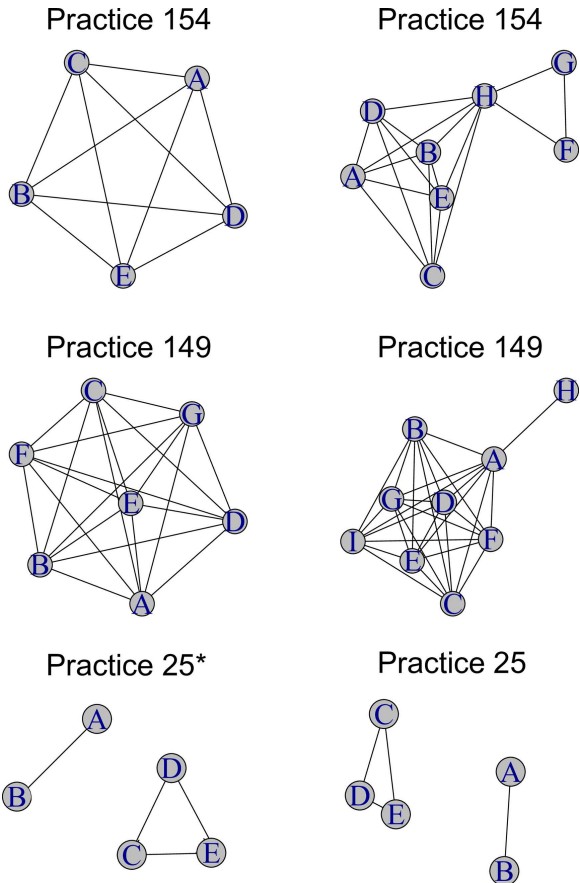

**Fig 1. Clinic structures based on administrative data with clinic-level identifiers (right side) and the comparable network-identified practice based on shared clients (left side).** Shared patient network (left) is created through prescription records and connections are defined as physicians prescribed to at least one shared patient. Clinic network (right) is identified through physician-clinic connection in the Canadian Addiction Treatment Centre (CATC) administrative records. Physicians are labelled to the unique practice. Edge thickness is the log of the shared clients plus 1. Vertex size is the log of the physicians' client load plus 1. *Practice 25 is over two different network-identified clinics as they cannot be grouped without shared clients.

We altered the tie definition to assess concordance among all tie classifications with the community detection algorithm set to modularity maximization with two iterations and physician-clinic connection defined at the 15% level (Table 2). Overall agreement and negative agreement were unchanged at 0.96 and 0.98, respectively, across all definitions. Three different tie definitions featured the highest positive agreement of 0.49. When trying to achieve the best balance among the four validity measures, the solution with high specificity (0.98 (0.98, 0.98)), high NPV (0.98 (0.97, 0.98)), medium PPV (0.54 (0.52, 0.57)) and medium sensitivity (0.45 (0.43, 0.47)) had two different tie definitions with the same result: Relative percentage defined as at the top 80% of both physicians shared clients or an absolute threshold of at least one client. Gwet's $AC_1$ estimates for both definitions were 0.96 (0.95, 0.96). Further, these were the highest performing set of all 350 versions of the detection algorithms, physician-clinic connection and tie definitions (S1 **and** S2 Fig and S3 Table). Other algorithms with higher PPV had low or near zero estimates for sensitivity or only medium NPV.

## Discussion

Using a previously validated methodology [26,27,36], we compared 350 different algorithms to validate the identification of OAT clinics from administrative data by creating prescriber networks from shared clients. Using a relative percentage to define network ties with two or three recursive iterations of the modularity maximization algorithm were identified as

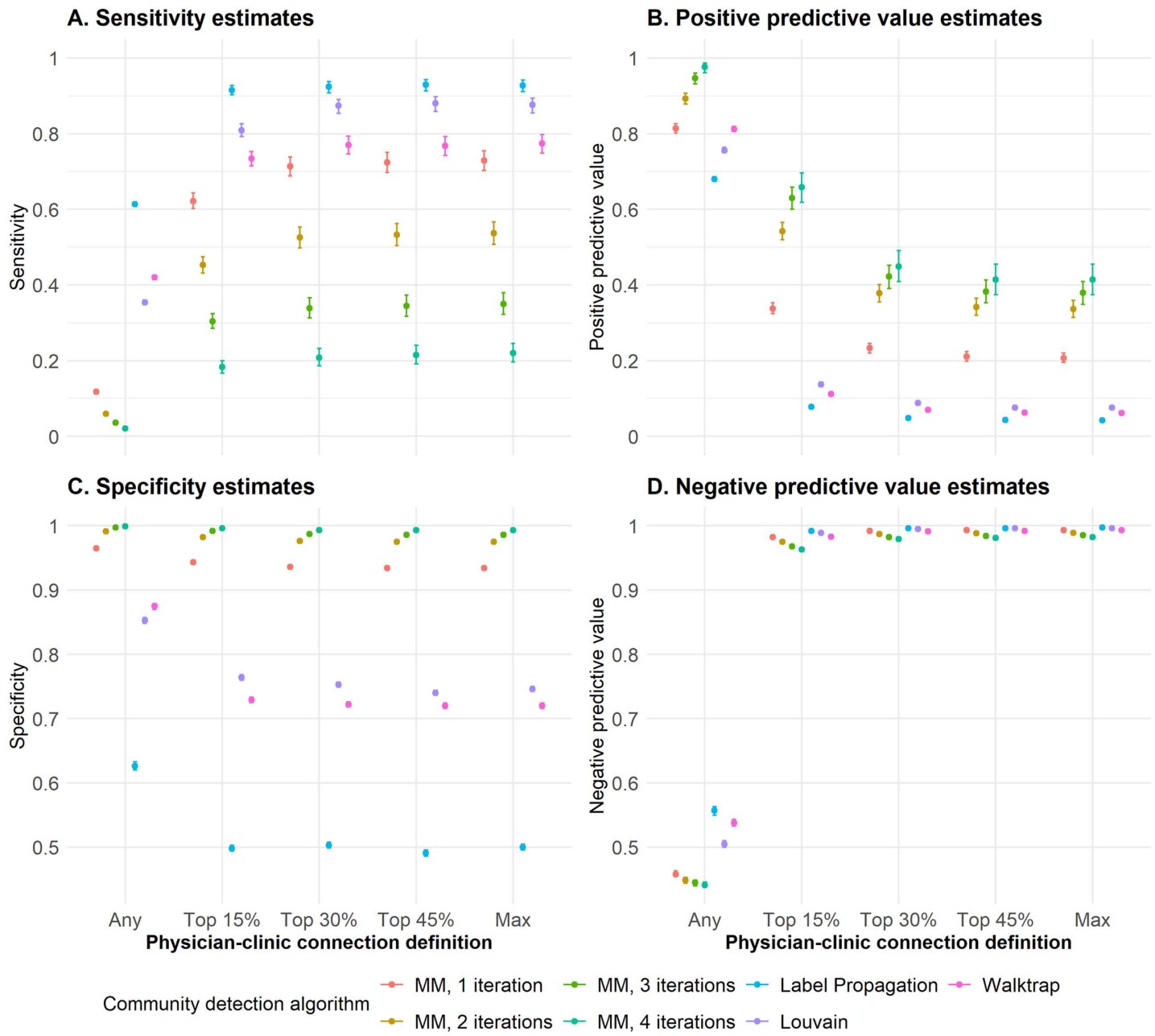

**Fig 2. Concordance statistics for different community detection algorithms and physician-clinic connection definition to identify the gold-standard administrative clinics.** Abbreviations: MM – Modularity maximization.

the best performing strategies for constructing network-defined clinics due to high specificity and NPV and moderate PPV and sensitivity. The misclassification from the algorithm occurred when physicians had low client loads or comparably high client loads across multiple clinics.

Generally, use of administrative databases for evaluation and quality improvement has limitations, particularly with missing information and the lack of identifying indicators for clinical sites and referral networks. These limitations are exceptionally relevant for clinical OUD management in the fentanyl-era, when rapidly accessible specialty addictions

**Table 2. Concordance between physician pairs' assigned network identified community and the gold-standard administrative identified clinic.**

| | Concordance statistics | | | | Validity statistics | | | |
|---|---|---|---|---|---|---|---|---|
| | Overall agree | Positive agree | Negative agree | Gwet's AC$_1$ (95% CI) | Sensitivity | Specificity | Positive predic- tive value | Negative predic- tive value |
| Network tie definition: Absolute number of shared clients | | | | | | | | |
| 1 | 0.96 | 0.49 | 0.98 | 0.96 (0.95, 0.96) | 0.45 (0.43, 0.47) | 0.98 (0.98, 0.98) | 0.54 (0.52, 0.57) | 0.98 (0.97, 0.98) |
| 2 | 0.96 | 0.46 | 0.98 | 0.96 (0.95, 0.96) | 0.39 (0.37, 0.41) | 0.98 (0.98, 0.99) | 0.55 (0.53, 0.58) | 0.97 (0.97, 0.97) |
| 5 | 0.96 | 0.40 | 0.98 | 0.96 (0.95, 0.96) | 0.31 (0.29, 0.33) | 0.99 (0.99, 0.99) | 0.55 (0.52, 0.57) | 0.97 (0.97, 0.97) |
| 10 | 0.96 | 0.37 | 0.98 | 0.96 (0.95, 0.96) | 0.28 (0.26, 0.30) | 0.99 (0.99, 0.99) | 0.57 (0.54, 0.60) | 0.97 (0.97, 0.97) |
| 20 | 0.96 | 0.31 | 0.98 | 0.96 (0.95, 0.96) | 0.21 (0.20, 0.23) | 0.99 (0.99, 0.99) | 0.56 (0.53, 0.60) | 0.97 (0.96, 0.97) |
| **Shared episode of OAT care** | 0.96 | 0.48 | 0.98 | 0.95 (0.95, 0.96) | 0.44 (0.42, 0.46) | 0.98 (0.98, 0.98) | 0.53 (0.51, 0.56) | 0.97 (0.97, 0.98) |
| Network tie definition: Relative % of each physicians' ties | | | | | | | | |
| 20% | 0.96 | 0.47 | 0.98 | 0.95 (0.95, 0.96) | 0.43 (0.41, 0.45) | 0.98 (0.98, 0.98) | 0.51 (0.48, 0.53) | 0.97 (0.97, 0.98) |
| 40% | 0.96 | 0.49 | 0.98 | 0.95 (0.95, 0.96) | 0.45 (0.43, 0.47) | 0.98 (0.98, 0.98) | 0.53 (0.50, 0.55) | 0.98 (0.97, 0.98) |
| 60% | 0.96 | 0.48 | 0.98 | 0.95 (0.95, 0.96) | 0.45 (0.43, 0.47) | 0.98 (0.98, 0.98) | 0.52 (0.50, 0.55) | 0.98 (0.97, 0.98) |
| 80% | 0.96 | 0.49 | 0.98 | 0.96 (0.95, 0.96) | 0.45 (0.43, 0.47) | 0.98 (0.98, 0.98) | 0.54 (0.52, 0.57) | 0.98 (0.97, 0.98) |

Presented results set the 'true' clinic to be a clinic a physician has at least 15% of the total patients at, and under the modularity maximization algorithm with at least 2 iterations. Results where these options were varied are in Fig 1 and **Table A3** of the supplementary appendix.

clinics [41] and collaborative care [1,4] have been introduced as one component to combat the declines in treatment retention and persistently high rates of overdose death among people who use drugs [8,42–44]. By validating the identification of clinics, we provide an opportunity to distinguish and evaluate clinical caseloads and clinic-level per- formance among databases that lack clinic-level identifiers, thus improving the use of administrative health records for ongoing evaluation and quality improvement particularly for opioid use disorder. Additionally in other clinical areas, care coordination, measured through indicators such as care density and centrality, has shown to have an impact on rates of hospitalization, health care costs, as well as on the referral and prescribing practices of physicians [5,28,29,45]. A previous study in the United States analyzed insurance claims to demonstrate that patients receiving care from doctors with higher levels of shared patients (i.e., higher care density) had significantly lower total costs and rates of hospitalization [28]. Otherwise, another study in the United States applied network analysis to demon- strate that higher care density is associated with a reduction in repeated co-prescription of interacting drugs by multiple providers [29]. Centrality, which describes the importance or influence of a node within a network, is another indicator of coordination [46]. For example, Barnett et al. compared the relative ratio of primary care physician (PCP) centrality to other physicians' centrality to quantify the average centrality of PCPs compared to other physicians in a network and found that higher "centrality" of primary care providers within constructed hospital networks was associated with fewer medical specialist visits, as well as lower spending on imaging and tests [45]. Validation of network-constructed clinics may therefore have broader implications and applicability in programmatic and health system evaluation.

Identifying facilities with administrative data otherwise provides an opportunity for enhanced control of confounding, which may be otherwise unmeasured in comparative effectiveness studies, when there is a lack of clinic-level identifiers. A previous study that identified predictors of selection into OAT had estimated individual and prescriber-level impacts with residual variance between prescribers [47]. This unexplained variance could be related to clinic-level characteristics that were unable to be determined previously. Furthermore, measures of prescribing preferences are commonly used in comparative effectiveness studies using instrumental variable analysis to address unmeasured confounding by indication

[48–52]. A 2021 systematic review of preference-based instrumental variable designs found that of 185 articles, 40% had implemented a facility-level measure of prescribing preference [50], highlighting another important use to identifying clinics with administrative data that lacks facility-level identifiers. Additionally, as physicians' peers' prescribing habits has been shown to impact treatment uptake and selection of medications [53–55], these outcomes could be improved by identifying the impact of clinic level prescribing practices.

The results from our chosen 'best-performing' settings had an estimated PPV of 0.54 and sensitivity of 0.45 which is comparable to the study by Kuo et al on US Medicare clients with similar sensitivity, however slightly lower PPVs [27]. Specifically, when Kuo et al defined a tie as at least 14 shared patient they had nearly the same sensitivity. However, we had much higher estimates for specificity and NPV. This may be explained by the administrative data having a higher proportion of physicians not practicing at the same clinics because they are easily identified if they do not share patients. Another previous study had similar results with a shared patient threshold of 3, resulting in a PPV of 0.60, sensitivity of 0.57, and specificity of 0.77, however their NPV was lower at 0.76 and their goal was to identify referral relationships, not team-based clinics [25]. Overall, our results align with other studies and confirm administrative databases can be used to identify physician relationships and clinics.

Prior studies building physician networks used an absolute shared patient threshold utilizing Medicare from the United States [25,27,36]. However, we found that when network ties were defined based on relative percentage thresholds the results were concordant among all of the percentage-based definitions. As such, and consistent with the findings and recommendations of Landon et al [26], our results suggest using relative thresholds over absolute shared patient thresholds for network construction to accommodate diverse clinical contexts with varying patient loads and case complexities. Another study by Landon et al suggested similar, as absolute thresholds will vary for different specialties [26]. While Landon et al did not report their results and comparisons, they concluded that defining a connection based on a relative threshold of the top 20% of ties was the best option [26]. Other studies evaluating prescriber networks defined connections based on episodes of care [56]. We found that episodes of care also had similar results to relative thresholds, suggesting that both definitions create valid prescriber networks.

Overall, our study found that Walktrap and Louvain algorithms underperformed compared to the modularity maximization algorithm. While there is no precedent in the prescriber networks literature that evaluated the use of different community detection algorithms, other complex network studies have evaluated differences in detection algorithms [57–61]. There is not a single algorithm that prevails as the optimal algorithm for community detection among different types of networks (i.e., small and large networks) and performance measures. For example, the Louvain algorithm provided the best fit in two studies [57,61], while the modularity maximization algorithm was superior in Berardo de Sousa et al [58]. Our modularity maximization method was different, such that we followed Stein et al's approach of recursive application to find smaller communities. After the third iteration, we found the minimal increase in PPV resulted in larger decrease of sensitivity, suggesting that higher iterations are unnecessary. This aligns with Stein et al, as they found the best split for minimum community sizes with minimum number of total communities was 3.3, aligning with our results that second or third iteration was best [34].

Our study was not without limitations. First, as physicians treated clients at multiple OAT clinics in Ontario, the network-identified communities will necessarily result in a degree of misclassification as a physician can only be assigned to a single community. To address this concern, we used multiple definitions for physician-clinic connection. In other settings and provinces physicians often work in multiple clinics, and we found the algorithm captures true groupings from clinics with at least 15% of the physicians' client-load. The algorithm may not be reliable at detecting clinics which exclusively or primarily employ prescribers with a small number of clients relative to the physicians' overall client-load. Future analyses should employ other sensitivity analyses, such as different detection algorithms and connection definitions to ensure clinic identification is consistent. While physicians often work in multiple clinics, our methodology required prescribers to be assigned to only one clinic. Algorithms to detect overlapping communities within weighted networks exist, such

as the Speaker-listener Label Propagation Algorithm [62], however they have not been used in prior prescriber network literature and are not widely implemented in software. Second, for the express purpose of defining OAT clinics, we focus our only on OAT prescription records. It is possible that some OAT physicians could also be working at non-OAT clinics and sharing non-OAT patients which we would not capture. The methodology can be extended in other applications to capture any prescriptions, or any records of care provision, between shared clients. We note however the unique benefits of using OAT prescription records, given requirements for frequent (weekly or biweekly) prescription renewals. We note that the method was validated to identify primary care offices in Texas using outpatient care records from US Medicare data [27], and communities within 51 different hospital referral regions using inpatient and outpatient care records from US Medicare data [26]. Elsewhere it has be implemented to identify hospital referral regions in England using hospital admission records to measure infection transfers between hospitals [63].

This analysis examined OAT prescriber network characteristics with the aim of validating the use of administrative data for identifying OAT prescriber networks. Identifying OAT prescriber networks from health administrative data can help determine the impact of physician networks on OAT outcomes and provide recommendations to improve collaborative care.

## Supporting Information

**S1 Table. Characteristics of the 350 different networks created each calendar year.**
(DOCX)

**S2 Table. Summary of characteristics for network-identified clinics and administrative identified clinics between 2013 and 2019.**
(DOCX)

**S3 Table. Concordance between physician pairs' assigned network identified community and the gold-standard administrative identified clinic for different algorithm settings.**
(DOCX)

**S1 Fig. Sensitivity and positive predictive value estimates.**
(DOCX)

**S2 Fig. Specificity and negative predictive value estimates.**
(DOCX)

## Author contributions

**Conceptualization:** Megan Kurz, Kristen A. Morin, Emanuel Krebs, Bohdan Nosyk.

**Data curation:** Mark Tatangelo, Kristen A. Morin.

**Formal analysis:** Megan Kurz.

**Funding acquisition:** Bohdan Nosyk.

**Methodology:** Megan Kurz.

**Supervision:** Bohdan Nosyk.

**Visualization:** Megan Kurz.

**Writing – original draft:** Megan Kurz, Kristen A. Morin, Michelle Zanette, Emanuel Krebs.

**Writing – review & editing:** Kristen A. Morin, Michelle Zanette, David C Marsh, Bohdan Nosyk.

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
