## [Decision Letter · Decision Letter 0]

15 Jan 2025

PONE-D-24-47217Identifying opioid agonist treatment prescriber networks from health administrative data: a validation study.PLOS ONE

Dear Dr. Nosyk,

Thank you for submitting your manuscript to PLOS ONE. After careful consideration, we feel that it has merit but does not fully meet PLOS ONE’s publication criteria as it currently stands. Therefore, we invite you to submit a revised version of the manuscript that addresses the points raised by reviewers. In particular, there are concerns expressed by reviewers regarding methodology. 

We look forward to receiving your revised manuscript.

Kind regards,

Marianna Mazza

Academic Editor

PLOS ONE

“This work was funded by the Canadian Institutes of Health Research award number no. PJT-190265 and National Institutes of Health/National Institute on Drug Abuse award no. R01-DA050629.”

3. In the online submission form you indicate that your data is not available for proprietary reasons and have provided a contact point for accessing this data. Please note that your current contact point is a co-author on this manuscript. According to our Data Policy, the contact point must not be an author on the manuscript and must be an institutional contact, ideally not an individual. Please revise your data statement to a non-author institutional point of contact, such as a data access or ethics committee, and send this to us via return email. Please also include contact information for the third party organization, and please include the full citation of where the data can be found.

Reviewers' comments:

Reviewer's Responses to Questions

**Comments to the Author**

1. Is the manuscript technically sound, and do the data support the conclusions?

Reviewer #1: Yes

Reviewer #2: Yes

2. Has the statistical analysis been performed appropriately and rigorously? 

Reviewer #1: Yes

Reviewer #2: Yes

3. Have the authors made all data underlying the findings in their manuscript fully available?

Reviewer #1: No

Reviewer #2: Yes

4. Is the manuscript presented in an intelligible fashion and written in standard English?

Reviewer #1: Yes

Reviewer #2: Yes

5. Review Comments to the Author

Reviewer #1: The present manuscript leverages the health administrative data to construct networks among the prescribers and detect communities among them. The authors further validate the results with the ground truth of shared facilities and show that the network analyses can be used to identify the patterns in prescribing behaviors.

I think the paper is pretty solid and easy to follow. One suggestion I have is that I think it would be beneficial if the authors could provide more technical details. For instance, the authors mention that they have performed various robustness checks using different methods to construct 250 different networks. Please describe the characteristics of these networks, such as their number of nodes and number of edges.

Also, the time range of the data is not very clear to me. In the results section, it appears that health administrative data from seven years was used to create different versions of the networks. However, the ground truth data was only from one year. I think the authors need to justify these choices and also test the impact of using data from different time ranges to create networks.

Reviewer #2: I appreciate the authors' time and effort in studying OAT prescriber networks. This type of analysis is rarely conducted, and I commend the authors for addressing such an under-explored area. However, a key concern is the practical implications of the findings. What can we do with the results of this analysis? While the paper includes some discussion on this, I found the arguments to be somewhat underdeveloped and not entirely convincing.

Another important point is the justification for the chosen statistical methods. For instance, while modularity maximization is a widely used approach for identifying subgroups, it would be helpful to explain why this method was preferred over other, potentially more advanced alternatives.

Additionally, the definitions of certain evaluation measures could be clarified. For example, readers without a strong statistical background may not be familiar with terms such as Positive Predictive Value (PPV) or Negative Predictive Value (NPV). Providing clearer definitions or brief explanations would improve accessibility and understanding for a broader audience.

6. PLOS authors have the option to publish the peer review history of their article (what does this mean? ). If published, this will include your full peer review and any attached files.

**Do you want your identity to be public for this peer review?** For information about this choice, including consent withdrawal, please see our Privacy Policy .

Reviewer #1: No

Reviewer #2: No

---

## [Author Response · Author response to Decision Letter 1]

11 Mar 2025

Reviewers' comments:

Reviewer's Responses to Questions

Comments to the Author

Reviewer #1: The present manuscript leverages the health administrative data to construct networks among the prescribers and detect communities among them. The authors further validate the results with the ground truth of shared facilities and show that the network analyses can be used to identify the patterns in prescribing behaviors.

I think the paper is pretty solid and easy to follow. One suggestion I have is that I think it would be beneficial if the authors could provide more technical details. For instance, the authors mention that they have performed various robustness checks using different methods to construct 250 different networks. Please describe the characteristics of these networks, such as their number of nodes and number of edges.

Response: To describe the characteristics across each different network definition, we have added an additional table (S1 Table) as supplementary material. The table summarizes characteristics of the networks for each calendar year.

Also, the time range of the data is not very clear to me. In the results section, it appears that health administrative data from seven years was used to create different versions of the networks. However, the ground truth data was only from one year. I think the authors need to justify these choices and also test the impact of using data from different time ranges to create networks.

Response: As each network was created on an annual basis, and we have added an additional table (S2 Table) as supplementary material that describes the characteristics compared to the ground truth for each calendar year.

Reviewer #2: I appreciate the authors' time and effort in studying OAT prescriber networks. This type of analysis is rarely conducted, and I commend the authors for addressing such an under-explored area. However, a key concern is the practical implications of the findings. What can we do with the results of this analysis? While the paper includes some discussion on this, I found the arguments to be somewhat underdeveloped and not entirely convincing.

Response: We have edited the discussion to highlight more on how this validation can be used with administrative data for opioid use disorder in particular.

(page 14, paragraph 2, line 255): “Generally, use of administrative databases for evaluation and quality improvement has limitations, particularly with missing information and the lack of identifying indicators for clinical sites and referral networks. These limitations are exceptionally relevant for clinical OUD management in the fentanyl-era, when rapidly accessible specialty addictions clinics(40) and collaborative care(1, 4) have been introduced as one component to combat the declines in treatment retention and persistently high rates of overdose death among people who use drugs(8, 41-43). By validating the identification of clinics, we provide an opportunity to distinguish and evaluate clinical caseloads and clinic-level performance among databases that lack clinic-level identifiers, thus improving the use of administrative health records for ongoing evaluation and quality improvement particularly for opioid use disorder. Additionally in other clinical areas, …”

(page 15, paragraph 2, line 282): “Identifying facilities with administrative data otherwise provides an opportunity for enhanced control of confounding, which may be otherwise unmeasured in comparative effectiveness studies, when there is a lack of clinic-level identifiers. …”

Another important point is the justification for the chosen statistical methods. For instance, while modularity maximization is a widely used approach for identifying subgroups, it would be helpful to explain why this method was preferred over other, potentially more advanced alternatives.

Response: We previously tested two different community detection alternatives, including: Walktrap and Louvain and have made this clearer. While Louvain also applies modularity maximization, Walktrap does not. However, we added the Label Propagation algorithm to test an additional community detection alternative that does not optimize modularity.

(page 9, paragraph 1, line 167): “In addition to the modularity maximization method to identify disjoint communities, we compared the Walktrap(38) and Louvain(39) methods to test different identification algorithms … We also compared the Label Propagation(40) algorithm, although it has not been used in community detection studies among prescribers.”

(page 11, paragraph 2, line 213): “Results from the Walktrap, Louvain, and Label Propagation algorithms had lower specificity and PPV compared to the modularity maximization algorithm … Label Propagation had the highest sensitivity, with the drawbacks of the lowest PPV and specificity.”

Additionally, the definitions of certain evaluation measures could be clarified. For example, readers without a strong statistical background may not be familiar with terms such as Positive Predictive Value (PPV) or Negative Predictive Value (NPV). Providing clearer definitions or brief explanations would improve accessibility and understanding for a broader audience.

Response: We have added a basic descriptive of each term, relative to their meaning for this analysis.

(page 8, paragraph 2, line 144): “We also calculated the sensitivity (probability for a physician pair to be grouped in a network-identified community together when they are attached to the same clinic), specificity (probability for a physician pair to be in separate network-identified communities when they do not practice at the same clinics), positive predictive value (PPV; probability of a physician pair practicing at the same clinic when they are grouped in a network-identified community) and negative predictive value (NPV; probability of a physician pair practicing at different clinics when they are in different network-identified communities).”

---

## [Editor Report · Decision Letter 1]

16 Mar 2025

Identifying opioid agonist treatment prescriber networks from health administrative data: a validation study.

PONE-D-24-47217R1

Dear Dr. Nosyk,

We’re pleased to inform you that your manuscript has been judged scientifically suitable for publication and will be formally accepted for publication once it meets all outstanding technical requirements.

Kind regards,

Marianna Mazza

Academic Editor

PLOS ONE
---

## [Editor Report · Acceptance letter]

PONE-D-24-47217R1

PLOS ONE

Dear Dr. Nosyk,

I'm pleased to inform you that your manuscript has been deemed suitable for publication in PLOS ONE. Congratulations! Your manuscript is now being handed over to our production team.

Kind regards,

on behalf of

Dr. Marianna Mazza

Academic Editor

PLOS ONE